# Understanding Grokking: Insights from Neural Network Robustness

## Abstract

Recently, an interesting phenomenon called grokking has gained much attention, where generalization occurs long after the models have initially overfitted the training data. We try to understand this seemingly strange phenomenon through the robustness of the neural network. From a robustness perspective, we show that the usually observed decreasing of $l_2$ weight norm of the neural network is theoretically connected to the occurrence of grokking. Therefore, we propose to use perturbation-based methods to enhance robustness and speed up the generalization process. Furthermore, we show that the speed-up of generalization when using our proposed method can be explained by learning the commutative law, a necessary condition when the model groks on the test dataset. In addition, we empirically observe that $l_2$ norm correlates with grokking on the test data not in a timely way and then proposes new metrics based on robustness that correlate better with the grokking phenomenon.

## 1 Introduction

The generalization of overparameterized neural networks has been a fascinating topic in the machine learning community, challenging classical learning theory intuitions. Recently, researchers have discovered a interesting phenomenon called grokking, initially observed in training on unconventional Algorithmic datasets (Power et al., 2022), and later found in traditional tasks such as image classification (Liu et al., 2022b). Grokking refers to the unexpected generalization of learning tasks that occurs long after the models have initially overfitted the training data. This phenomenon has garnered increasing attention due to its resemblance to "phase transition" (Nanda et al., 2023). Since its initial report (Power et al., 2022), this phenomenon has garnered numerous explanations from various aspects (Nanda et al., 2023; Liu et al., 2022b; Merrill et al., 2023; Barak et al., 2022; Davies et al., 2023; Thilak et al., 2022; Gromov, 2023; Notsawo Jr et al., 2023; Varma et al., 2023).

One popular explanation for grokking is based on the decay of the network's $l_2$ weight norm (Liu et al., 2022b; Nanda et al., 2023) where weight decay is usually employed. From Figure 1, we can see that when the test accuracy quickly increases, the weight norm quickly drops correspondingly. Then a natural question arises: Does the happening of grokking closely related to the decay of weight norm? *Theoretically*, we can show that the decay of weight norm is indeed connected with grokking. Informally speaking, grokking occurs when the network's robustness increases to a certain level. When the weight norm decreases, the robustness increases, explaining why the decay of the weight norm is usually observed before grokking happens. Motivated by the above theoretical observations, we then explore the application of perturbation-based training to enhance robustness, thus accelerating generalization.

To deeper investigate the inner mechanism of the proposed perturbation-based training strategy. For one of the tasks (on Modulo Addition Dataset) in Figure 1, comprehending the commutative law of modulo addition is necessary for grokking on this task. One may expect the model to first comprehend this "easy" (necessary condition) task and then understand the full task of modulo addition. However, we discover an unusual observation during standard training on the modulo addition dataset (Power et al., 2022): the model fails to recognize the commutative law of modulo addition before fully understanding the task, which goes against our intuition. Leveraging this finding, we can explain the effectiveness of our perturbation-based approach by verifying that it successfully learns commutative law on the training dataset before grokking.

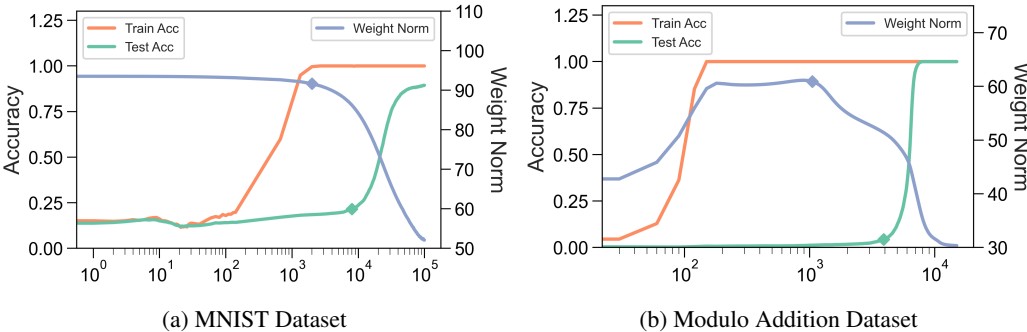

(a) MNIST Dataset  (b) Modulo Addition Dataset

Figure 1: Typical grokking cases on MNIST and Modulo Addition Dataset: Grokking phenomena refer to sudden and unforeseen enhancements of test accuracy far beyond the point when training accuracy reaches 100%.

On the other hand, recall that in Figure 1, we have observed that there is no *simultaneous* change between the weight norm decay and grokking on the test dataset. In light of this observation, we propose the utilization of novel metrics derived from both robustness theory and information theory. By incorporating these innovative metrics, our findings indicate a strong correlation with the phenomenon of grokking. This suggests that these newly introduced metrics might provide deeper insights into the underlying mechanisms driving grokking, thereby offering a more comprehensive understanding of the process.

Our contributions can be summarized as follows:

- We theoretically explain why decay of $l_2$ weight norm is closely linked with grokking through the robustness of the network.

- Motivated by the intuition that perturbation training may improve robustness, we use perturbation-based training to speed up generalization.

- We find a surprising fact that standard training on Modulo Addition Dataset fails to learn commutative law before grokking, which contradicts intuition. We further use this to explain the success of our new perturbation-based strategy.

- Borrowing ideas from robustness and information theory, we introduce new metrics that correlate better with the grokking process.

## 2 RELATED WORKS

**Grokking.**  Grokking, originally introduced by Power et al. (2022), refers to the intriguing phenomenon observed in small transformers training on algorithmic tasks. It has been noted that, after an extended period of training, test accuracy experiences a sudden improvement, transitioning from near-random to perfect understanding. Several explanations have been proposed to shed light on this very unusual phenomenon. Nanda et al. (2023) demonstrate that Fourier analysis can provide insights into understanding grokking. In contrast, Barak et al. (2022) hypothesize that the network steadily progresses towards generalization rather than making abrupt leaps. Furthermore, smaller instances of grokking have been explored by Liu et al. (2022a), unveiling distinct learning phases. Importantly, Thilak et al. (2022) argue that grokking can manifest without explicit regularization, proposing the existence of an implicit regularizer known as the slingshot mechanism. Furthermore, Liu et al. (2022b) demonstrate that grokking is not limited to modulo addition tasks but can occur in more conventional tasks as well.

**Information theory in understanding neural networks.**  Information theory has long provided a valuable framework for understanding the interplay between probability and information. It has been used to understand the inner workings of neural networks (Tishby et al., 2000; Tishby & Zaslavsky, 2015), trying to make the mechanism of neural network operations more "white-box". As the computation burden of information-theoretic quantities is usually high, a recent departure from

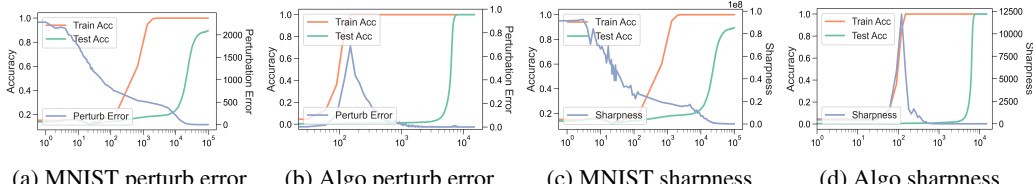

(a) MNIST perturb error     (b) Algo perturb error     (c) MNIST sharpness     (d) Algo sharpness

Figure 2: Robustness and sharpness on MNIST and Modulo Addition training dataset, we use Algo to indicate Modulo Addition dataset to save the space for image caption. (Set the deviation of Gaussian perturbation as $\sigma = 0.04$). We plot the perturbation error as a robustness indicator in (a) and (b), a **smaller** perturbation error means more robust.

traditional information theory has emerged, shifting the focus towards generalizing it for measuring relationships between matrices (Bach, 2022; Skean et al., 2023; Zhang et al., 2023b;a). Tan et al. (2023) apply matrix mutual information and joint entropy to self-supervised learning.

## 3 PROBLEM SETTING AND NOTATIONS

In this paper, we mainly consider the setting of supervised classification, the most common case in grokking (Liu et al., 2022b; Nanda et al., 2023). Let $\mathcal{D}_{train} = \{(\mathbf{x}_i, y_i)\}_{i=1}^n$ be the training dataset, and $W$ denote the weight parameters of the neural network, and $f(\mathbf{x}_i, W)$ be the output of neural network on sample $\mathbf{x}_i$. The empirical training loss is defined as $\frac{1}{n}\sum_{i=1}^n \ell(f(\mathbf{x}_i, W), y_i)$, where $\ell$ is a loss function such as mean squared error (MSE) loss and cross-entropy loss. In this paper, we consider two canonical grokking settings for experiments: One is the MNIST image classification task introduced by (Liu et al., 2022b); another is the modulo addition dataset (Power et al., 2022).

To train our model on the MNIST Dataset, we follow the setup of (Liu et al., 2022b). We adopt a width-200 depth-3 ReLU MLP architecture and employ MSE loss. The optimization is performed using the AdamW optimizer with a learning rate of 0.001, and we utilize a batch size of 200 during the training process. Our experiments on the Modulo Addition Dataset follow the setup of (Nanda et al., 2023). We train transformers to perform addition modulo $P$, where the input format is "$ab =$" (sometimes we call it "$a + b =$", which is more intuitive), with $a$ and $b$ encoded as $P$-dimensional one-hot vectors. The output $c$ is read above the special token "$=$". We use $P = 113$ and a one-layer ReLU transformer. The token embeddings have a dimension of $d = 128$, with learned positional embeddings. The model includes 4 attention heads of dimension $\frac{d}{4} = 32$ and an MLP with $k = 512$ hidden units. For evaluation, we use a training dataset comprising 30% of all possible inputs pairs $(a, b)$, training loss is the standard cross-entropy loss for classification.

Due to the space limitation, all the proofs are deferred to Appendix A.

## 4 GROKKING: A ROBUSTNESS PERSPECTIVE

### 4.1 UNDERSTANDING THE ROLE OF $l_2$ WEIGHT NORM DECAY IN GROKKING

From Figure 1, we can see that when the network starts to grok, there is usually a sharp decrease in the network's $l_2$ weight norm. The decay of weight norm can be partially explained by applying the weight decay, but one may wonder if the decay of weight norm has some theoretical explanations with the appearance of grokking. In the following, we will show that the decay of $l_2$ weight norm is indeed connected with the grokking phenomenon from a theoretical viewpoint.

For the simplicity of theoretical analysis, we consider the MSE loss in this section, e.g., the empirical risk can be written as $\mathcal{L}(W) := \frac{1}{2n}\sum_{i=1}^n \|f(\mathbf{x}_i, W) - \text{onehot}(y_i)\|_2^2$.

Let $W^* = (W_1^*, W_2^*)$ be an interpolation solution, i.e., $\mathcal{L}(W^*) = 0$, namely, $f(\mathbf{x}_i, W^*) = \text{onehot}(y_i)$ on the entire training dataset. Define sharpness of the loss function at $W^*$ as the sum of the eigenvalues of the Hessian $\nabla^2\mathcal{L}(W^*)$, i.e., $S(W^*) := \text{Tr}(\nabla^2\mathcal{L}(W^*))$.

We will first present a lemma which is mainly adapted from (Ma & Ying, 2021). This lemma describes the relationship of weight norm, sharpness, and the robustness of a neural network.

**Lemma 4.1.** *Suppose $W^*$ is an interpolation solution, then the following inequality holds:*

$$\frac{1}{n}\sum_{i=1}^{n}\|\nabla_{\mathbf{x}}f(\mathbf{x}_i, W^*)\|_F^2 \leq \frac{\|W^*\|_F^2}{\min_i \|\mathbf{x}_i\|_2^2}S(W^*).$$

To see how the robustness of network behaves during training, we will try to directly evaluate the perturbation error (a **smaller** perturbation error means more robust): $\sum_{i=1}^{n}\|f(\mathbf{x}_i + \Delta_i, W) - f(\mathbf{x}_i, W)\|_F^2$, where $\Delta_i \sim \mathcal{N}(0, \sigma^2\mathbf{I})$ is Gaussian noise. We plot the perturbation error in Figure 2 (a) and (b) and find it decreases after train accuracy reaches $100\%$. As lemma 4.1 also involves the sharpness. We plot the sharpness in Figure 2 (c) and (d) and find it continuously decreases during training. Then as the weight norm also decreases, lemma 4.1 shows that the perturbation error will decrease which fits empirical results.

Then we use lemma 4.1 to understand grokking, *informally* we show: Under some assumptions, we can successfully classify all the test samples in a certain "diameter" of training samples, lemma 4.1 is used to determine the expression of "diameter". Formally, we have theorem 4.2. This theorem can be seen as showing why the decrease of $l_2$ weight norm is beneficial for grokking.

**Theorem 4.2.** *Suppose $W^*$ is a interpolation solution and the gradient of $f(\mathbf{x}, W^*)$ is L-Lipschitz about $\mathbf{x}$. Assume at least $\delta$-fraction of test data has a train dataset neighbor whose distance is at most $\epsilon(W^*)$ and has a same label with the associated test data, where $\epsilon(W^*) = \min\{1, \frac{1}{2(\sqrt{\frac{n}{\min_i \|\mathbf{x}_i\|_2^2}}\|W^*\|_F^2 S(W^*) + L)}\}$. Then the test accuracy will be at least $\delta$.*

From theorem 4.2, it is clear that when $l_2$ norm decays, the distance threshold $\epsilon(W^*)$ increases thus the ratio of test samples who has a train dataset neighbor of distance at most $\epsilon(W^*)$ increases. This assumption on the distance between training and testing data is intuitive as they are usually treated as generated from the same underlying distribution, making the samples with the same label very close. Then it is clear that the test accuracy increases, which means that $l_2$ weight norm decay is indeed beneficial for the generalization on the test dataset.

The name grokking consists of two things. One is that it generalizes on test data eventually, which we have discussed previously. Another is that when it generalizes, the accuracy increases sharply, mimicking a sort of "phase transition" phenomenon. We will further analyze the latter in detail.

Define the distance function to the training dataset as follows:

$$d(\mathbf{x}, \mathcal{D}_{train}) = \min_{\mathbf{y} \in \mathcal{D}_{train}} \|\mathbf{x} - \mathbf{y}\|_2. \tag{1}$$

Motivated by theorem 4.2, we can define the neighbouring probability of test data distribution $P_{test}$ as: $P(r) = \mathbb{E}_{\mathbf{X} \sim P_{test}}\mathbb{I}(d(\mathbf{X}, \mathcal{D}_{train}) \leq r) = \Pr(d(\mathbf{X}, \mathcal{D}_{train}) \leq r)$.

Then theorem 4.2 shows that the test accuracy will be at least $P(\epsilon(W^*))$. Therefore, we have the following corollary.

**Corollary 4.3.** *Under the same assumption of theorem 4.2. Suppose $L \geq \frac{1}{2}$, $\|\mathbf{x}_i\|_2 = 1$ and $\|W^*\|_F^2 S(W^*) = \frac{\max^2\{a - b\log_{10}(\text{train-steps}), 0\}}{4n}$. The the test accuracy will be at least $P(\frac{1}{2L + \max\{a - b\log_{10}(\text{train-steps}), 0\}})$.*

Assume $Y \sim \mathcal{N}(0, \mu)$ is a normal distribution of variance $\mu$, as the distance function is always non-negative. A very natural form of $P(r)$ will be $P(r) = \Pr(-r \leq Y \leq r)$. We will see that this assumption will make us simulate the "phase transition" very closely.

Take $L = \frac{1}{2}$, $\mu = \frac{1}{100}$, $a = 1925$ and $b = 500$. We take the total training steps as $15000$, which is the same as the Modulo Addition dataset. We plot the predicted accuracy given by corollary 4.3 in Figure 3. It is interesting to see that the predicted accuracy closely matches that of the real accuracy, it may provide a possible explanation for the "phase transition" phenomenon in grokking.

For more discussions on $l_1$ weight norm, please see Appendix A.1.

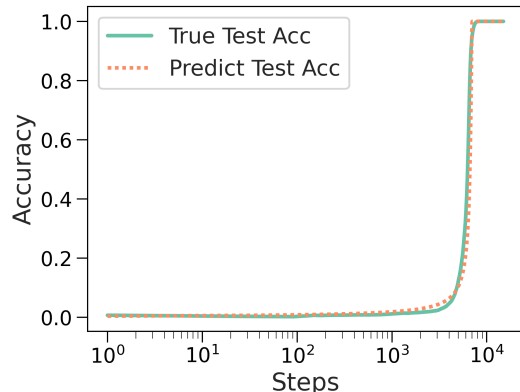

Figure 3: The predicted accuracy matches the real test accuracy on the Modulo Addition dataset.

## 4.2 A DIRECT DEGROKKING STRATEGY BASED ON ROBUSTNESS

Our previous findings suggest that the robustness of a neural network plays a significant role in the process of grokking. In light of this, we need an approach to enhance the robustness of neural networks directly and thus speed up the generalization process (which we termed "degrokking").

To achieve this, we have designed a method that introduces controlled perturbations to the input data during the training process. By doing so, we aim to induce the neural network to increase its resilience to variations and uncertainties in the input data, thus increasing the robustness of the network. This approach allows us to maintain the integrity of the initial training while making minimum modifications. We made minimum changes to the initial training by only adding $\Delta \sim \mathcal{N}(0, \sigma^2 \mathbf{I})$ to the input. As grokking has an extremely unequal speed of convergence on training and testing dataset compared to non-grokking standard training cases, adding a constant strength perturbation to the training input may result in over-perturbation. To better consider the training process, we adaptively update $\sigma = \max(\lambda_1(1 - \text{train acc}), \lambda_2)$, this makes the perturbation strength transits much more smoothly. Formally, the training objective function is as follows:

$$\frac{1}{n} \sum_{i=1}^{n} \ell(f(\mathbf{x}_i + \Delta_i, W), y_i), \tag{2}$$

where $\mathbf{x}_i$ is the image in MNIST and the input token embedding in the Modulo Addition Dataset.

We plot the curves in Figure 4 (a) and (b), it is clear that this perturbation-based strategy speeds up the generalization. On MNIST, $\lambda_1 = 0.06$ and $\lambda_2 = 0.03$. On Modulo Addition Dataset, $\lambda_1 = 0.5$ and $\lambda_2 = 0.4$. We attribute the staircase-like curve on the algorithmic dataset to "first-memorize, then generalize" and we left the detailed study as future work. Compared to (Liu et al., 2022b), our strategy can be seen as a traditional training technique. It may also explain why grokking is not common in usual settings, where data augmentations can be seen as adding a sort of perturbation to the input.

## 5 A CLOSER LOOK AT GROKKING ON MODULO ADDITION DATASET

### 5.1 NECESSARY CONDITION FROM GROUP THEORY

As the task is modulo adding, from group theory, it shall obey the commutative law once it actually learns the general addition rule.

Formally, the commutative law in an abelian group is

$$a + b = b + a, \tag{3}$$

where $+$ is the binary group operation considered.

In Figure 5, we plot the accuracy of when the prediction of $a + b$ is equal to that of $b + a$ on samples $(a, b)$s. Note the prediction is based on the maximal index of the "$a + b =$" ("$b + a =$") logits. We

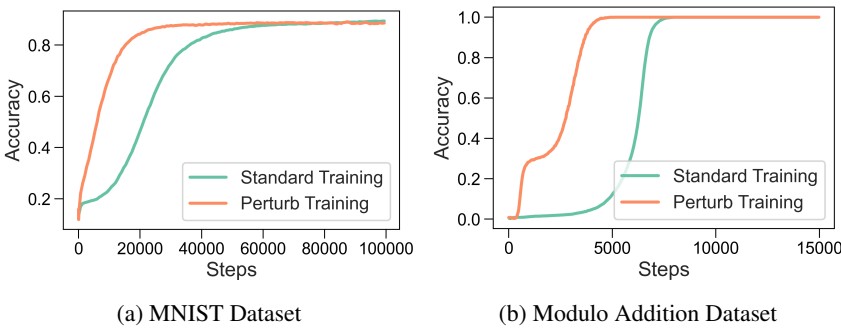

(a) MNIST Dataset  (b) Modulo Addition Dataset

Figure 4: Our perturbed training strategy speeds up generalization ("degrokking").

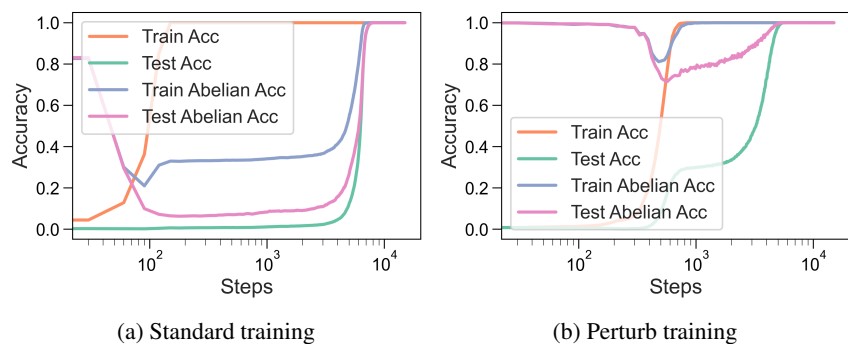

(a) Standard training  (b) Perturb training

Figure 5: Abelian test shows that perturb training will help model comprehend commutative rule.

call this "abelian test". Surprisingly, from Figure 5, it is clear that the standard training process does not commutative rule on the training data until it groks. However, the perturbed training strategy comprehends commutative rule on the training data right after the training accuracy reaches 100%.

In Figure 6, we further add the logits level MSE loss of training samples "$a + b =$" and "$b + a =$" as regularizer on the initial loss. We call this strategy abelian degrok, in the experiment we set the regularization coefficient of the regularizer to 100. We find it speeds up training and performs even better than the perturb-based strategy. Therefore, combining the observations in Figures 6 and 5, we think that the effectiveness of our perturb-based strategy may be explained by learning the commutative law.

## 5.2 DOES THE ROBUSTNESS-BASED DEGROKKING STRATEGY CHANGE THE REPRESENTATION?

At first glance, the robustness (perturbation)-based degrokking strategy made a minimum change to the initial training. Thus, one may doubt whether it has learned a similar representation as the initial one. In Figure 5 (a) and (b), we can see that the abelian test accuracy of both standard and perturb training is stable after it first reaches a 100% test accuracy. But as our design of abelian accuracy only implicitly involves representation, we need more detailed quantities to see whether the representation of these two types of training coincides or not.

In Figure 7, we plot the distance of the logits between "$a + b =$" and "$b + a =$" on samples $(a, b)$s. This quantity depends directly on the representation and we call this "Abelian test on logits". Figure 7 (a) shows that the logits distance of standard training is stable once it reaches 0, but (b) has a chaotic behavior when the test accuracy reaches 100%. It clearly shows that the representations obtained by standard and perturb training are different, as the dynamic differs a lot.

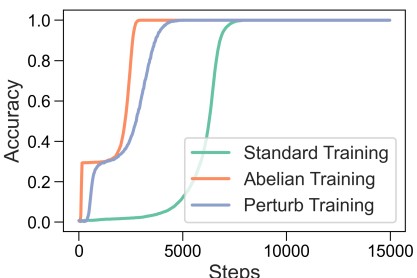

Figure 6: Abelian degrok strategy.

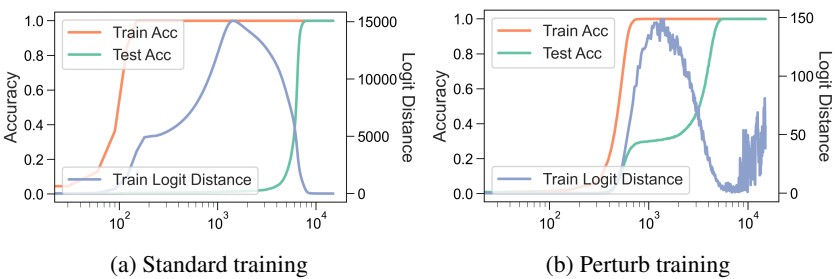

(a) Standard training          (b) Perturb training

Figure 7: Abelian test on logits.

## 6   NEW METRICS THAT CORRELATE BETTER TO TEST ACCURACY

We have found that the decay of $l_2$ weight norm is beneficial for grokking on the test dataset (theorem 4.2). But from Figure 1, we can see that decay of weight norm usually happens *before* grokking happens. One may wonder can we propose new metrics that better correlate with the grokking process? We answer this question in the affirmative by borrowing ideas from robustness theory and combined with intuitions from information theory. More discussions can be found in Appendix B.

### 6.1   METRICS BASED ON ROBUSTNESS AND INFORMATION THEORY

#### 6.1.1   MATRIX-BASED INFORMATION THEORETIC QUANTITIES

In this section, we will briefly summarize some (matrix-based) information-theoretic quantities that we will use (Skean et al., 2023). These quantities are more computationally tractable than the traditional estimations based on representation distribution (Paninski, 2003).

**Definition 6.1** ($\alpha$-order matrix entropy). Suppose a positive semi-definite matrix $\mathbf{R} \in \mathbb{R}^{n \times n}$ which $\mathbf{R}(i,i) = 1$ for every $i = 1, \cdots, n$ and $\alpha > 0$. The $\alpha$-order (Rényi) entropy for matrix $\mathbf{R}$ is defined as follows:

$$\mathrm{H}_\alpha\left(\mathbf{R}\right) = \frac{1}{1-\alpha} \log\left[\mathrm{tr}\left(\left(\frac{1}{n}\mathbf{R}\right)^\alpha\right)\right],$$

where $\mathbf{R}^\alpha$ is the matrix power.

The case of $\alpha = 1$ recovers the von Neumann (matrix) entropy, i.e.,

$$\mathrm{H}_1\left(\mathbf{R}\right) = -\mathrm{tr}\left(\frac{1}{n}\mathbf{R}\log\frac{1}{n}\mathbf{R}\right).$$

In this paper, if not stated otherwise, we will always use $\alpha = 1$. Using the definition of matrix entropy, we can define matrix mutual information as follows.

**Definition 6.2** (Matrix mutual information). Suppose positive semi-definite matrices $\mathbf{R}_1, \mathbf{R}_2 \in \mathbb{R}^{n \times n}$ which $\mathbf{R}_1(i,i) = \mathbf{R}_2(i,i) = 1$ for every $i = 1, \cdots, n$. $\alpha$ is a positive real number. The $\alpha$-order matrix mutual information for matrix $\mathbf{R}_1$ and $\mathbf{R}_2$ is defined as follows:

$$\mathrm{I}_\alpha(\mathbf{R}_1; \mathbf{R}_2) = \mathrm{H}_\alpha(\mathbf{R}_1) + \mathrm{H}_\alpha(\mathbf{R}_2) - \mathrm{H}_\alpha(\mathbf{R}_1 \odot \mathbf{R}_2).$$

### 6.1.2 METRICS

As we want to have metrics that correlate more closely with the grokking process, motivated by the use of information theory to explain neural network learning (Tishby & Zaslavsky, 2015), we want to use metrics that are information-theoretically understandable. But as information-theoretic quantities are usually hard to compute exactly, we find that matrix information-theoretic quantities may be better choices as they are simpler to compute.

Note when computing matrix information-theoretic quantities, we need to satisfy its requirement (i.e. an all $1$ diagonal and positive semi-definiteness). A very easy construction of a matrix satisfying these requirements is the ($l_2$ normalized) feature gram matrix. Note features are evaluated on specific datasets, during training we only have access to the training dataset. As we have discussed the usage of robustness in explaining grokking, we would like the features calculated based on a perturbed training dataset.

We will give the definite of our metrics "perturbed mutual information" based on the intuition discussed above. Informally, perturbed mutual information is defined as the (feature $l_2$ normalized gram matrix) mutual information of the input and output layers on dataset $\mathcal{D}_{train}$, when input is perturbed by $\Delta \sim \mathcal{N}(0, \sigma^2 \mathbf{I})$. **Denote $W = (W_1, W_2)$, where $W_1$ is the first layer of the neural network.**

**Definition 6.3** (Perturb Mutual Information). Suppose the perturbation is sampled as $\Delta = (\Delta_i)_{i=1}^n \sim \mathcal{N}(0, \sigma^2 \mathbf{I})$. Denote $\mathbf{Z}_1(\Delta) = [\mathbf{z}_1^{(1)} \cdots \mathbf{z}_n^{(1)}]$, where $\mathbf{z}_i^{(1)} = \frac{f(\mathbf{x}_i + \Delta_i, W_1)}{\|f(\mathbf{x}_i + \Delta_i, W_1)\|}$. Also denote $\mathbf{Z}_2(\Delta) = [\mathbf{z}_1^{(2)} \cdots \mathbf{z}_n^{(2)}]$, where $\mathbf{z}_i^{(2)} = \frac{f(\mathbf{x}_i + \Delta_i, W)}{\|f(\mathbf{x}_i + \Delta_i, W)\|}$. Define $\mathbf{G}_1(\Delta) = \mathbf{Z}_1^T(\Delta)\mathbf{Z}_1(\Delta)$ and $\mathbf{G}_2(\Delta) = \mathbf{Z}_2^T(\Delta)\mathbf{Z}_2(\Delta)$. The perturb mutual information is defined as:

$$\text{PMI}(f, \sigma, \alpha) = \mathbb{E}_{\Delta \sim \mathcal{N}(0, \sigma^2 \mathbf{I})}\, \text{I}_\alpha(\mathbf{G}_1(\Delta), \mathbf{G}_2(\Delta)). \tag{4}$$

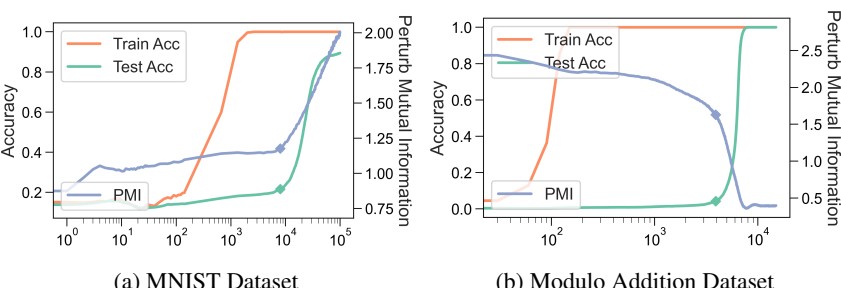

(a) MNIST Dataset      (b) Modulo Addition Dataset

Figure 8: Perturb mutual information on the training dataset.

Similarly, we define the perturb entropy metric as follows.

**Definition 6.4** (Perturb Entropy). Suppose the perturbation is sampled as $\Delta = (\Delta_i)_{i=1}^n \sim \mathcal{N}(0, \sigma^2 \mathbf{I})$. Denote $\mathbf{Z}(\Delta) = [\mathbf{z}_1 \cdots \mathbf{z}_n]$, where $\mathbf{z}_i = \frac{f(\mathbf{x}_i + \Delta_i, W)}{\|f(\mathbf{x}_i + \Delta_i, W)\|}$. Define $\mathbf{G}(\Delta) = \mathbf{Z}^T(\Delta)\mathbf{Z}(\Delta)$. The perturb entropy is defined as:

$$\text{PE}(f, \sigma, \alpha) = \mathbb{E}_{\Delta \sim \mathcal{N}(0, \sigma^2 \mathbf{I})}\, \text{H}_\alpha(\mathbf{G}(\Delta)). \tag{5}$$

From the definition of perturb mutual information and entropy, we can see it has a relatively high computation overhead. Therefore, we will make a few approximations to accelerate the computing. The parameter $\alpha$ is usually set as $1$ as this closely aligns with the classical Shannon entropy. As the number of samples is relatively high, we instead calculate the quantity within each batch and the average among batches. Note for ease of calculation, we sample $\Delta$ only once.

We plot the perturb mutual information (PMI) in Figure 8 (a) and (b), it is clear that perturb mutual information changes sharply at the time test accuracy undergoes a quick increase. On MNIST, $\sigma = 0.1$. On Modulo Addition Dataset, $\sigma = 0.4$.

We plot the perturb entropy (PE) in Figure 9 (a) and (b), it is clear that perturb entropy changes sharply at the time test accuracy undergoes a quick increase.

From the above observations, we find that the proposed two metrics PMI and PE change sharply *at the time* of grokking. This makes them better indicators for grokking than the $l_2$ weight norm.

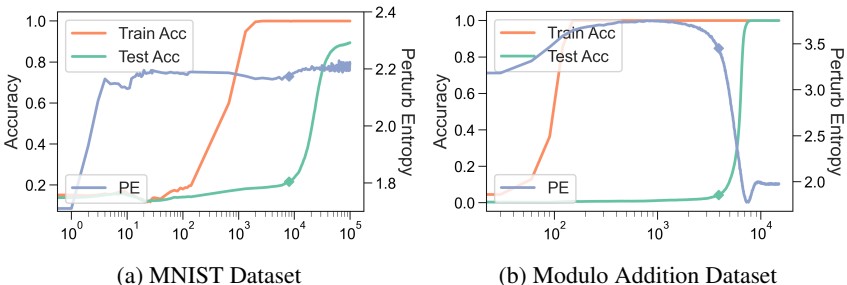

(a) MNIST Dataset        (b) Modulo Addition Dataset

Figure 9: Perturb entropy on the training dataset.

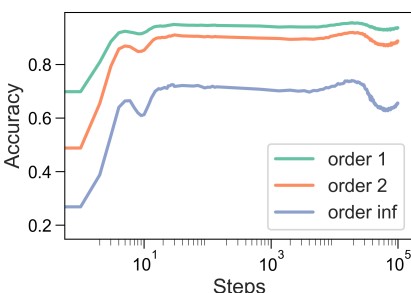

Figure 10: Ablation on entropy order.

## 7 ABLATION STUDY

In Figure 10, we adjust the parameter $\alpha$ which is employed in the calculation of matrix entropy. The primary goal of varying $\alpha$ is to demonstrate that the trends identified in this paper are not significantly affected by the specific choice of $\alpha$. This is to ensure that our findings are robust across different settings of this parameter. Additionally, to examine the discrepancies between the training and testing datasets, we perform these calculations on the unperturbed MNIST test dataset. The results indicate that the observed tendencies are stable not only across various values of $\alpha$ but also between the training and testing datasets. This reinforces the robustness of our approach and findings in both parameter variation and dataset differentiation.

## 8 CONCLUSION

In this paper, we explore the phenomenon of grokking through the lens of neural network robustness. Our investigation reveals that the decay of the $l_2$ weight norm in grokking experiments is not just an empirical observation but is theoretically linked to the grokking process. Based on these insights, we have proposed perturbation-based methods aimed at accelerating the generalization speed of neural networks.

Furthermore, we observed that during standard training on the Modulo Addition Dataset, there is an initial absence of fundamental group operation learning before the appearance of grokking. Interestingly, our proposed method to speed up generalization can be understood by the learning of the commutative law, an essential component for successful generalization on the test dataset during training.

Additionally, we introduced new metrics derived from robustness considerations that demonstrate a strong correlation with the occurrence of grokking. These metrics not only provide a quantitative measure of the learning process but also enhance our understanding of the dynamics underpinning the grokking phenomenon. Future work may include trying to find the grokking phenomenon on bigger models and test our understanding.

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

# Appendix

## A  DETAILED PROOFS

**Theorem A.1.** *Suppose $W^*$ is a interpolation solution and the gradient of $f(\mathbf{x}, W^*)$ is $L$-Lipschitz about $\mathbf{x}$. Assume at least $\delta$-fraction of test data has a train dataset neighbor whose distance is at most $\epsilon(W^*)$ and has a same label with the associated test data, where $\epsilon(W^*) = \min\{1, \frac{1}{2(\sqrt{\frac{n}{\min_i \|\mathbf{x}_i\|_2^2}}\|W^*\|_F^2 S(W^*)+L)}\}$. Then the test accuracy will be at least $\delta$.*

*Proof.* For a test data $\mathbf{x}'_i$, denote its training data neighbour as $\mathbf{x}_i$, then

$$
\begin{aligned}
\|f(\mathbf{x}_i, W^*) - f(\mathbf{x}'_i, W^*)\| &\le \|\mathbf{x}_i - \mathbf{x}'_i\|\|\nabla_{\mathbf{x}} f(\hat{\mathbf{x}}, W^*)\| \\
&\le \|\mathbf{x}_i - \mathbf{x}'_i\|(\|\nabla_{\mathbf{x}} f(\mathbf{x}_i, W^*)\| + \|\nabla_{\mathbf{x}} f(\hat{\mathbf{x}}, W^*) - \nabla_{\mathbf{x}} f(\mathbf{x}_i, W^*)\|) \\
&\le \|\mathbf{x}_i - \mathbf{x}'_i\|(\|\nabla_{\mathbf{x}} f(\mathbf{x}_i, W^*)\| + L\|\mathbf{x}_i - \mathbf{x}'_i\|) \\
&\le \|\mathbf{x}_i - \mathbf{x}'_i\|(\|\nabla_{\mathbf{x}} f(\mathbf{x}_i, W^*)\| + L) \\
&\le \|\mathbf{x}_i - \mathbf{x}'_i\|(\sqrt{\frac{n}{\min_i \|\mathbf{x}_i\|_2^2}}\|W^*\|_F^2 S(W^*) + L) \le \frac{1}{2}.
\end{aligned}
$$

Thus the difference between $f(\mathbf{x}_i, W^*)$ and $f(\mathbf{x}'_i, W^*)$ is bounded by $\frac{1}{2}$, as $W^*$ is the interpolating solution. Taking the usual argmax decision rule will make $\mathbf{x}'_i$ successfully classified. The assumption on Lipschitz is standard in the literature, loosening this assumption is also an interesting future work. □

### A.1  BEHAVIOR OF $l_1$ WEIGHT NORM

We will then analyze another type of regularization technique, for example, $l_1$ weight decay.

**Corollary A.2.** *Under the same assumption of theorem 4.2. Assume at least $\delta$-fraction of test data has a train dataset neighbor whose distance is at most $\hat{\epsilon}(W^*)$, where $\hat{\epsilon}(W^*) = \min\{1, \frac{1}{2(\sqrt{\frac{n}{\min_i \|\mathbf{x}_i\|_2^2}}\|W^*\|_1^2 S(W^*)+L)}\}$ and has a same label with the associated test data. Then the test accuracy will be at least $\delta$.*

*Proof.* Denote vec as the operation of reshaping the tensor to a column vector. Then define $w = \text{vec}(W^*)$, it is clear that $\|W^*\|_F^2 = \|w\|_2^2$ and $\|W^*\|_1 = \|w\|_1$. As $\|w\|_2^2 = \sum_i w_i^2 = \sum_i |w_i|^2 \le (\sum_i |w_i|)(\sum_i |w_i|) = \|w\|_1^2$, combined with theorem 4.2 the conclusion follows. □

The corollary A.2 can show that $l_1$ norm decay will also result in grokking. Žunkovič & Ilievski (2022) discuss that $l_1$ is better than $l_2$ decay in $1D$ exponential model, this can be understood by the proof of corollary A.2 as the decay of $l_1$ norm imposes more constraint than $l_2$ norm. We have also plotted the behavior of $l_1$ weight norm during training in Figure 11.

## B  MORE DISCUSSIONS ON ROBUSTNESS INSPIRED METRICS

### B.1  METRICS BASED ON ROBUSTNESS AND MODEL CONFIDENCE

Given a prediction vector $f(\mathbf{x}_i, W) = [f(\mathbf{x}_i, W)_1 \cdots f(\mathbf{x}_i, W)_K]^\top$ of the model on a sample $\mathbf{x}_i$ and denote $\hat{y}_i = \arg\max_j f(\mathbf{x}_i, W)_j$, one can construct a confidence score $s(\mathbf{x}_i, W) = \frac{|f(\mathbf{x}_i, W)_{\hat{y}_i}|}{\sum_{j=1}^K |f(\mathbf{x}_i, W)_j|}$ (MSE loss) or $s(\mathbf{x}_i, W) = \frac{e^{f(\mathbf{x}_i, W)_{\hat{y}_i}}}{\sum_{j=1}^K e^{f(\mathbf{x}_i, W)_j}}$ (cross-entropy loss). As we've discussed the use of robustness to explain grokking, we would like the confidence scores to be calculated using a perturbed training dataset.

We will then provide the definition of perturbation confidence. The metrics will based on hidden representations compared to the ones in section 6.1.2 that are based on the prediction logits.

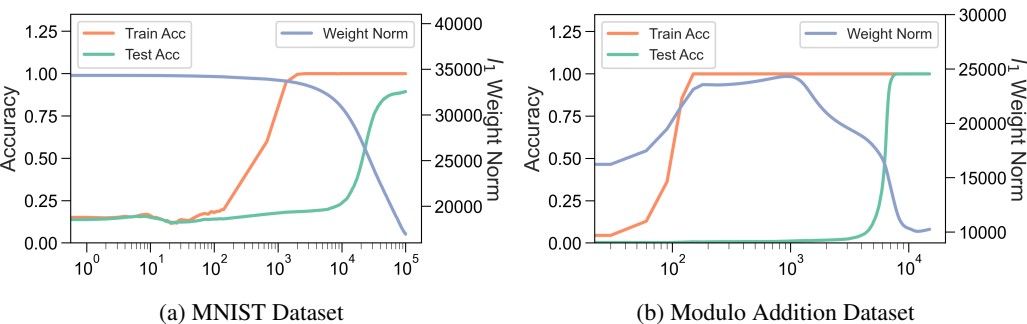

Figure 11: The behaviour of $l_1$ weight norm during training.

**Definition B.1** (Perturb Confidence). Given the dataset $\{(\mathbf{x}_i, y_i)\}_{i=1}^n$. The perturb confidence is defined as:

$$\mathrm{PC}(f, \sigma) = 100 \times \frac{1}{n} \sum_{i=1}^n \mathbb{E}_{\Delta_i \sim \mathcal{N}(0, \sigma^2 \mathbf{I})} s(\mathbf{x}_i + \Delta_i, W).$$

We plot the perturb confidence (PC) in Figure 12 (a) and (b), it is clear that the tendency of perturb confidence changes sharply at the time test accuracy undergoes a quick increase. On MNIST, the perturb confidence turns to a quick increase, and on the Modulo Addition Dataset, the perturb confidence turns into a decrease from the initial increasing tendency. On MNIST, $\sigma = 0.04$. On Modulo Addition Dataset, $\sigma = 0.1$.

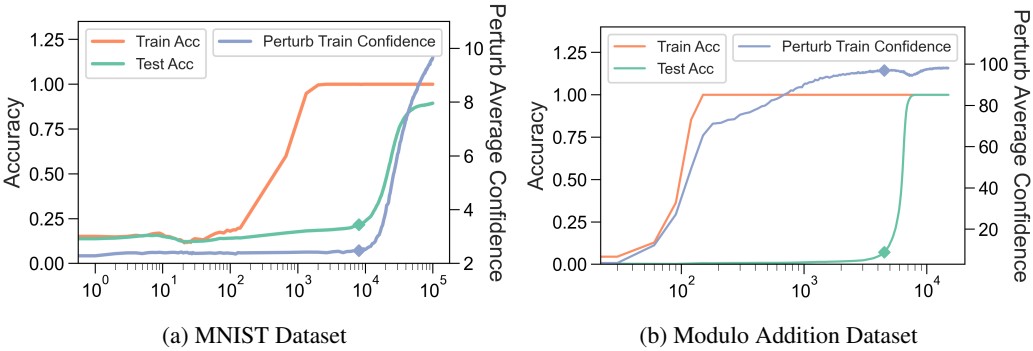

Figure 12: Perturb confidence on the training dataset.

### B.2 METRICS THAT MAY CORRELATE WITH THE APPEARANCE OF GROKKING

#### B.2.1 CONFIDENCE BASED

Since the metric PC shows a close correlation with the grokking process, it raises the question of whether this metric can predict the existence of grokking by monitoring the dynamics during the early stages of training. Inspired by the close connection between robustness and grokking, we have discovered that observing the differences in confidence between perturbed and unperturbed confidence may help us pinpoint which processes are likely to lead to grokking eventually.

**Definition B.2.** (Confidence Difference) $\mathrm{CD}(f, \sigma) = |\mathrm{PC}(f, \sigma) - \mathrm{PC}(f, 0)|$.

In Figure 13, one can see that after the training accuracy reaches $100\%$ and before grokking happens, the Confidence Difference (CD) will undergo a decreasing phase, meaning that the robustness of the network is enhanced during training. As robustness is closely linked with grokking, this decreasing signal of CD may be correlated with the appearance of grokking. All experiments can be conducted on a single Nvidia 2080Ti in less than 1 day.

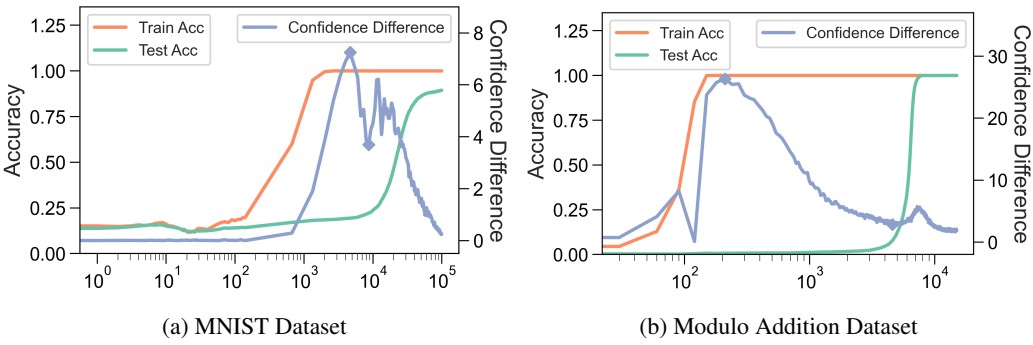

(a) MNIST Dataset                    (b) Modulo Addition Dataset

Figure 13: Grokking and confidence difference.

### B.2.2 INFORMATION THEORY BASED

As the metrics PMI and PE closely correlate with the grokking process. One may wonder can these metrics give intuitions of whether a learning process will eventually grok or not by observing the dynamics in the *early* stage of training. Motivated by the fact that robustness is closely related to the grokking process, we find that the difference between perturbed and un-perturbed information-theoretic quantities may be correlated with the appearance of grokking.

**Definition B.3.** (Mutual Information Difference) $\text{MID}(f, \sigma, \alpha) = |\text{PMI}(f, \sigma, \alpha) - \text{PMI}(f, 0, \alpha)|$.

**Definition B.4.** (Entropy Difference) $\text{ED}(f, \sigma, \alpha) = |\text{PE}(f, \sigma, \alpha) - \text{PE}(f, 0, \alpha)|$.

As we are caring about "predicting", this means that we should focus only on the early stage training. Therefore we are considering the training process *before* the training accuracy reaches $100\%$. We set $\alpha$ and $\sigma$ just as section 6. In Figure 14 and 15, we can see when both MID or ED have a phase of consistent decreasing. Additionally, on MNIST, the MID and ED are very small throughout the whole training process.

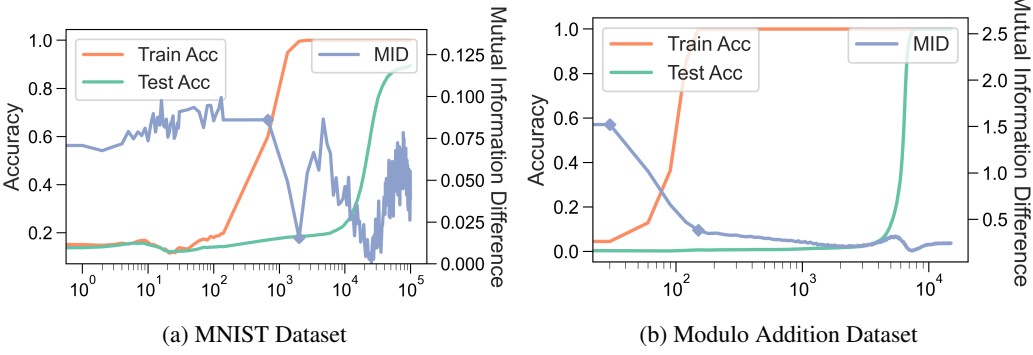

(a) MNIST Dataset                    (b) Modulo Addition Dataset

Figure 14: Grokking and mutual information difference.

From the experiments, we thus hypothesize that grokking happens when MID and ED are very small or undergo a sharp decreasing phase *before* training accuracy reaches $100\%$. This has an intuitive explanation: As we made small perturbations in the input, the resulting perturbation in representations will reflect the robustness of the network. Thus MID and ED will reflect the robustness of the network and we have shown robustness closely correlates with grokking. *Note MID and ED are calculated with hidden representation, it usually has a decreasing signal much earlier than CD, which is based on the predicted logits and may be not that sensitive with the inner structure of the network.*

We will then prove that the above indexes (MID and ED) may indicate grokking from a robustness viewpoint. In this subsection, we assume $\alpha = 2$ in the matrix information quantities to simplify the proof. This implication is based on the fact that we find that matrix information quantities have tendencies robust to the choice of $\alpha$ and we do ablation studies in section 7.

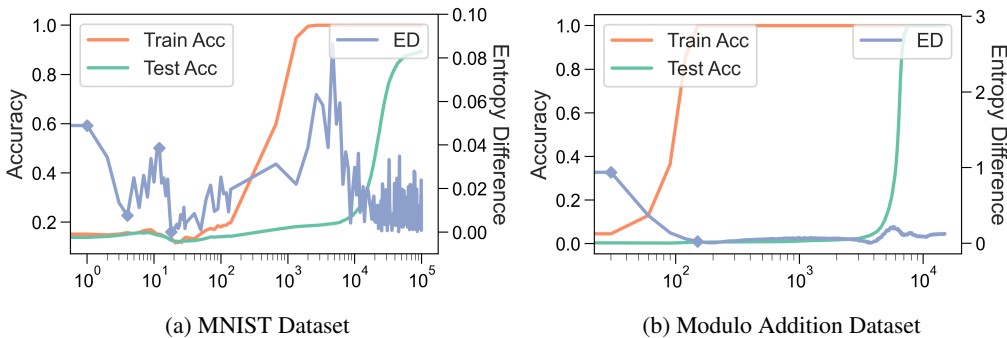

(a) MNIST Dataset       (b) Modulo Addition Dataset

Figure 15: Grokking and entropy difference.

The following two lemmas (Tan et al., 2023) show that mutual information and entropy can be transformed into norms, which is more related in the context of robustness.

**Proposition B.5.** $I_2(\mathbf{R}_1; \mathbf{R}_2) = 2\log n - \log\frac{||\mathbf{R}_1||_F^2 ||\mathbf{R}_2||_F^2}{||\mathbf{R}_1 \odot \mathbf{R}_2||_F^2}$, *where $n$ is the size of matrix $\mathbf{R}_1$ and $\mathbf{R}_2$ and $F$ is the Frobenius norm.*

**Proposition B.6.** *Suppose $\mathbf{K} \in \mathbb{R}^{n \times n}$. Then $H_2(\mathbf{K}) = 2\log n - \log ||\mathbf{K}||_F^2$, where $F$ is the Frobenius norm.*

We will show that the more robust a network is, the smaller the indexes (MID and ED) we defined above. We will discuss the small differences in the representations that will result in small perturbation of each element in the gram matrix, thus small distances in the Frobenius norm.

The following lemma shows that the perturbation of the gram matrix can be controlled by the perturbation of representations.

**Lemma B.7.** *Suppose vectors $\mathbf{a}_i$, $\mathbf{a}_j$ and $\mathbf{b}_i$, $\mathbf{b}_j$ are $l_2$ normalized, then $|\langle \mathbf{a}_i, \mathbf{a}_j \rangle - \langle \mathbf{b}_i, \mathbf{b}_j \rangle| \leq 4(\|\mathbf{a}_i - \mathbf{b}_i\| + \|\mathbf{a}_j - \mathbf{b}_j\|)$.*

*Proof.*

$$
\begin{aligned}
|\langle \mathbf{a}_i, \mathbf{a}_j \rangle - \langle \mathbf{b}_i, \mathbf{b}_j \rangle| &= |\|\mathbf{a}_i - \mathbf{a}_j\|^2 - \|\mathbf{b}_i - \mathbf{b}_j\|^2| \\
&= |(\|\mathbf{a}_i - \mathbf{a}_j\| + \|\mathbf{b}_i - \mathbf{b}_j\|)(\|\mathbf{a}_i - \mathbf{a}_j\| - \|\mathbf{b}_i - \mathbf{b}_j\|)| \\
&\leq 4|\|\mathbf{a}_i - \mathbf{a}_j\| - \|\mathbf{b}_i - \mathbf{b}_j\|| \\
&= 4|\|\mathbf{a}_i - \mathbf{b}_i + \mathbf{b}_j - \mathbf{a}_j + \mathbf{b}_i - \mathbf{b}_j\| - \|\mathbf{b}_i - \mathbf{b}_j\|| \\
&\leq 4(\|\mathbf{a}_i - \mathbf{b}_i\| + \|\mathbf{a}_j - \mathbf{b}_j\|).
\end{aligned}
$$

$\square$

Using lemmas B.6 and B.7, we can obtain the following bounds on the difference of entropy.

**Lemma B.8.** *Suppose $\mathbf{Z} = [\mathbf{z}_1 \cdots \mathbf{z}_n]$ and $\mathbf{Z}' = [\mathbf{z}'_1 \cdots \mathbf{z}'_n]$ have each of their columns $l_2$ normalized. Denote $\mathbf{G} = \mathbf{Z}^T \mathbf{Z}$ and $\mathbf{G}' = (\mathbf{Z}')^T \mathbf{Z}'$. Then we have the following inequality:*

$$
|H_2(\mathbf{G}) - H_2(\mathbf{G}')| \leq 8 \sum_{i=1}^{n} \|\mathbf{z}_i - \mathbf{z}'_i\|.
$$

*Proof.*

$$| \operatorname{H}_2(\mathbf{G}_1) - \operatorname{H}_2(\mathbf{G}_1')| = |\log \|\mathbf{G}_1\|_F^2 - \log \|\mathbf{G}_1'\|_F^2|$$

$$= |\log \frac{\|\mathbf{G}_1\|_F^2}{\|\mathbf{G}_1'\|_F^2}|$$

$$= \log(1 + \frac{|\|\mathbf{G}_1\|_F^2 - \|\mathbf{G}_1'\|_F^2|}{\min\{\|\mathbf{G}_1\|_F^2, \|\mathbf{G}_1'\|_F^2\}})$$

$$\leq \log(1 + \frac{|\|\mathbf{G}_1\|_F^2 - \|\mathbf{G}_1'\|_F^2|}{n})$$

$$\leq \frac{|\|\mathbf{G}_1\|_F^2 - \|\mathbf{G}_1'\|_F^2|}{n}$$

$$= \frac{|\sum_{i,j}\langle \mathbf{z}_i, \mathbf{z}_j \rangle^2 - \langle \mathbf{z}_i', \mathbf{z}_j' \rangle^2|}{n}$$

$$\leq \frac{\sum_{i,j} |(\langle \mathbf{z}_i, \mathbf{z}_j \rangle - \langle \mathbf{z}_i', \mathbf{z}_j' \rangle)(\langle \mathbf{z}_i, \mathbf{z}_j \rangle + \langle \mathbf{z}_i', \mathbf{z}_j' \rangle)|}{n}$$

$$\leq 2 \frac{\sum_{i,j} |\langle \mathbf{z}_i, \mathbf{z}_j \rangle - \langle \mathbf{z}_i', \mathbf{z}_j' \rangle|}{n}$$

$$\leq 8 \frac{\sum_{i \neq j} \|\mathbf{z}_i - \mathbf{z}_i'\| + \|\mathbf{z}_j - \mathbf{z}_j'\|}{n}$$

$$= 8 \sum_{i=1}^{n} \|\mathbf{z}_i - \mathbf{z}_i'\|.$$

$\square$

Then it is straightforward to obtain the following bound on ED using lemma B.8.

**Theorem B.9.** *The entropy difference can be bounded by the perturbation of network representation as follows:*

$$\operatorname{ED}(f, \sigma, 2)$$
$$\leq 8 \mathbb{E}_{\Delta \sim \mathcal{N}(0, \sigma^2 \mathbf{I})} \sum_{i=1}^{n} \|f(\mathbf{x}_i + \Delta_i, W) - f(\mathbf{x}_i, W)\|.$$

The calculation of mutual information involves Hadarmard product. The following lemma bounds the perturbation of Hadarmard product.

**Lemma B.10.** *If numbers $|a|$ $|a'|$ and $|b|$ $|b'|$ all less or equal than 1, then $|ab - a'b'| \leq |a - a'| + |b - b'|$.*

*Proof.* Note $ab - a'b' = (a - a')b + a'(b - b')$. Then the conclusion follows from the triangular inequality. $\square$

Using lemma B.5, B.7, B.8, and B.10, we can obtain the following bounds on the difference of mutual information.

**Lemma B.11.** *Suppose $\mathbf{Z}_1 = [\mathbf{z}_1^{(1)} \cdots \mathbf{z}_n^{(1)}]$, $\mathbf{Z}_2 = [\mathbf{z}_1^{(2)} \cdots \mathbf{z}_n^{(2)}]$ and $\mathbf{Z}_1' = [(\mathbf{z}_1')^{(1)} \cdots (\mathbf{z}_n')^{(1)}]$, $\mathbf{Z}_2' = [(\mathbf{z}_1')^{(2)} \cdots (\mathbf{z}_n')^{(2)}]$ have each of their columns $l_2$ normalized. Denote $\mathbf{G}_1 = \mathbf{Z}_1^T \mathbf{Z}_1$, $\mathbf{G}_2 = \mathbf{Z}_2^T \mathbf{Z}_2$ and $\mathbf{G}_1' = (\mathbf{Z}_1')^T \mathbf{Z}_1'$, $\mathbf{G}_2' = (\mathbf{Z}_2')^T \mathbf{Z}_2'$. Then we have the following inequality:*

$$| \operatorname{I}_2(\mathbf{G}_1, \mathbf{G}_2) - \operatorname{I}_2(\mathbf{G}_1', \mathbf{G}_2')| \leq 16 \sum_{j=1}^{2} \sum_{i=1}^{n} \|\mathbf{z}_i^{(j)} - (\mathbf{z}_i')^{(j)}\|.$$

*Proof.* Note $| \operatorname{I}_2(\mathbf{G}_1, \mathbf{G}_2) - \operatorname{I}_2(\mathbf{G}_1', \mathbf{G}_2')| = |(\operatorname{H}_2(\mathbf{G}_1) - \operatorname{H}_2(\mathbf{G}_1')) + (\operatorname{H}_2(\mathbf{G}_2) - \operatorname{H}_2(\mathbf{G}_2')) + (\operatorname{H}_2(\mathbf{G}_1' \odot \mathbf{G}_2') - \operatorname{H}_2(\mathbf{G}_1 \odot \mathbf{G}_2))| \leq | \operatorname{H}_2(\mathbf{G}_1) - \operatorname{H}_2(\mathbf{G}_1')| + | \operatorname{H}_2(\mathbf{G}_2) - \operatorname{H}_2(\mathbf{G}_2')| + | \operatorname{H}_2(\mathbf{G}_1' \odot \mathbf{G}_2') - \operatorname{H}_2(\mathbf{G}_1 \odot \mathbf{G}_2)| \leq | \operatorname{H}_2(\mathbf{G}_1' \odot \mathbf{G}_2') - \operatorname{H}_2(\mathbf{G}_1 \odot \mathbf{G}_2)| + 8 \sum_{j=1}^{2} \sum_{i=1}^{n} \|\mathbf{z}_i^{(j)} - (\mathbf{z}_i')^{(j)}\|.$

$$| \mathrm{H}_2(\mathbf{G}_1' \odot \mathbf{G}_2') - \mathrm{H}_2(\mathbf{G}_1 \odot \mathbf{G}_2)|$$

$$= | \log \|\mathbf{G}_1' \odot \mathbf{G}_2'\|_F^2 - \log \|\mathbf{G}_1 \odot \mathbf{G}_2\|_F^2 |$$

$$= | \log \frac{\|\mathbf{G}_1' \odot \mathbf{G}_2'\|_F^2}{\|\mathbf{G}_1 \odot \mathbf{G}_2\|_F^2} |$$

$$= \log(1 + \frac{|\|\mathbf{G}_1' \odot \mathbf{G}_2'\|_F^2 - \|\mathbf{G}_1 \odot \mathbf{G}_2\|_F^2|}{\min\{\|\mathbf{G}_1' \odot \mathbf{G}_2'\|_F^2, \|\mathbf{G}_1 \odot \mathbf{G}_2\|_F^2\}})$$

$$\leq \log(1 + \frac{|\|\mathbf{G}_1' \odot \mathbf{G}_2'\|_F^2 - \|\mathbf{G}_1 \odot \mathbf{G}_2\|_F^2|}{n})$$

$$\leq \frac{|\|\mathbf{G}_1' \odot \mathbf{G}_2'\|_F^2 - \|\mathbf{G}_1 \odot \mathbf{G}_2\|_F^2|}{n}$$

$$= \frac{|\sum_{i,j}(\mathbf{G}_1'(i,j)\mathbf{G}_2'(i,j))^2 - (\mathbf{G}_1(i,j)\mathbf{G}_2(i,j))^2|}{n}$$

$$\leq \frac{\sum_{i,j}|(\mathbf{G}_1'(i,j)\mathbf{G}_2'(i,j) - \mathbf{G}_1(i,j)\mathbf{G}_2(i,j))(\mathbf{G}_1'(i,j)\mathbf{G}_2'(i,j) + \mathbf{G}_1(i,j)\mathbf{G}_2(i,j))|}{n}$$

$$\leq 2\frac{\sum_{i,j}|\mathbf{G}_1'(i,j)\mathbf{G}_2'(i,j) - \mathbf{G}_1(i,j)\mathbf{G}_2(i,j)|}{n}$$

$$\leq 2\frac{\sum_{i,j}|\mathbf{G}_1'(i,j) - \mathbf{G}_1(i,j)| + |\mathbf{G}_2'(i,j) - \mathbf{G}_2(i,j)|}{n}$$

$$\leq 8\frac{\sum_{i\neq j}\|\mathbf{z}_i^{(1)} - (\mathbf{z}_i')^{(1)}\| + \|\mathbf{z}_j^{(1)} - (\mathbf{z}_j')^{(1)}\| + \|\mathbf{z}_i^{(2)} - (\mathbf{z}_i')^{(2)}\| + \|\mathbf{z}_j^{(2)} - (\mathbf{z}_j')^{(2)}\|}{n}$$

$$= 8\sum_{j=1}^{2}\sum_{i=1}^{n}\|\mathbf{z}_i^{(j)} - (\mathbf{z}_i')^{(j)}\|.$$

$\square$

It is interesting to see that if the input gram matrix is $\mathbf{I}$, then $\mathrm{I}_\alpha(\mathbf{I}, \mathbf{G}) = \mathrm{H}_\alpha(\mathbf{G})$. Note this is usually the case when considering raw pixel picture gram matrix, making mutual information a more "broad" quantity.

Then it is straightforward to obtain the following bound on MID using lemma B.11.

**Theorem B.12.** *The entropy difference can be bounded by the perturbation of network representation as follows:*

$$\mathrm{MID}(f, \sigma, 2) \leq 16\mathbb{E}_{\Delta \sim \mathcal{N}(0, \sigma^2 \mathbf{I})} \sum_{i=1}^{n}(\|f(\mathbf{x}_i + \Delta_i, W)$$
$$- f(\mathbf{x}_i, W)\| + \|f(\mathbf{x}_i + \Delta_i, W_1) - f(\mathbf{x}_i, W_1)\|),$$

*recall that $W_1$ is the first layer of the network.*

Theorem B.9 and B.12 clearly show that MID and ED have a close relationship with network robustness. Therefore making them good candidates that may be correlated with the appearance of grokking.

## C RELATION WITH INFORMATION BOTTLENECK

In Figure 16, we plot the matrix mutual information of perturbed logits and one-hot labels gram matrices on MNIST, which we termed PMI'. We find it is not as timely as PMI to grokking. Note PMI and PMI' can be seen as matrix information versions of the quantities used in information bottleneck (Tishby et al., 2000; Tishby & Zaslavsky, 2015). If we view input as variable $X$, logits as $Z$, and labels as $Y$. Then PMI is similar to $\mathrm{MI}(X, Z)$ and PMI' is similar to $\mathrm{MI}(Z, Y)$.

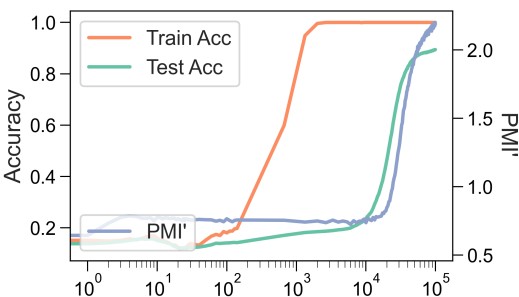

Figure 16: Perturb mutual information ablation.

