# OpenReview forum: "Understanding Grokking: Insights from Neural Network Robustness"
_ICLR.cc/2025/Conference — Submitted to ICLR 2025_

### Official Review · Reviewer_Nb9e · 2024-11-04

**Soundness:** 3
**Presentation:** 2
**Contribution:** 2
**Rating:** 5
**Confidence:** 2

**Summary:**

This paper investigates the phenomenon of "grokking" in neural network training, where models fully fit the training data but experience a prolonged period of low test accuracy before suddenly achieving high generalization performance. Through theoretical analysis and experiments, the authors reveal a relationship between grokking and the decay of the l2 weight norm, proposing a robustness-based "degrokking" strategy to accelerate grokking. Specifically, they demonstrate that the reduction in the l2 weight norm enhances model robustness, thus improving generalization to test data. The authors also introduce novel information-theory-based metrics that aim to better capture the grokking phenomenon, with empirical evidence supporting the effectiveness of these metrics.

**Strengths:**

1. The paper provides a new theoretical framework to explain the underlying mechanisms of grokking and proposes a robustness-based degrokking method, offering a fresh perspective on the study of grokking.
2. The authors design new metrics based on information theory that show potential in capturing grokking effectively.
3. The experiments on MNIST and the Modulo Addition dataset support the theoretical findings, demonstrating the relationship between l2 weight decay and grokking.

**Weaknesses:**

1. The study and experiments are based primarily on small-scale tasks and datasets, making it difficult to assess the generality of the theory in larger, more complex datasets or deeper networks.
2. Although the authors propose a perturbation strategy for degrokking, they do not clearly differentiate this approach from traditional data augmentation methods or discuss its unique advantages in accelerating generalization.

**Questions:**

1. While the authors demonstrate a relationship between l2 weight decay and grokking, the applicability of this theory to more complex datasets or large-scale models (e.g., modern deep neural networks) is not discussed. For instance, does this phenomenon persist when the dataset and task complexity increase? Could similar l2 weight decay behavior be observed on more complex tasks?
2. The proposed degrokking strategy introduces perturbations to enhance robustness, thereby accelerating generalization. How does this method fundamentally differ from traditional data augmentation techniques? If the goal is to accelerate generalization, could conventional data augmentation methods (e.g., random noise) achieve similar effects? Additionally, the authors’ theory appears to support the idea that data augmentation enhances generalization—would it be helpful to further clarify the relationship between the two?
3. Although the decay in the l2 weight norm is related to grokking, how does this phenomenon relate to the model’s learning dynamics and generalization ability? This relationship could be more complex in larger models. Can the authors’ theoretical explanation be extended to more diverse model architectures and tasks?

---

> ### Author Response · Authors · 2024-11-23
> **Rebuttal to Reviewer Nb9e**
>
> >W1 & Q1: The study and experiments are based primarily on small-scale tasks and datasets, making it difficult to assess the generality of the theory in larger, more complex datasets or deeper networks. While the authors demonstrate a relationship between l2 weight decay and grokking, the applicability of this theory to more complex datasets or large-scale models (e.g., modern deep neural networks) is not discussed. For instance, does this phenomenon persist when the dataset and task complexity increase? Could similar l2 weight decay behavior be observed on more complex tasks?
>
> A: The use of rather small-scale tasks and datasets in our study was influenced by the prevalence of such practices in previous literature, which all focus on small-scale settings. And we have already stated this limitation in our conclusion part.
>
>
> >W2 & Q2: Although the authors propose a perturbation strategy for degrokking, they do not clearly differentiate this approach from traditional data augmentation methods or discuss its unique advantages in accelerating generalization. The proposed degrokking strategy introduces perturbations to enhance robustness, thereby accelerating generalization. How does this method fundamentally differ from traditional data augmentation techniques? If the goal is to accelerate generalization, could conventional data augmentation methods (e.g., random noise) achieve similar effects? Additionally, the authors’ theory appears to support the idea that data augmentation enhances generalization—would it be helpful to further clarify the relationship between the two?
>
> A: There may be a misunderstanding. The goal of presenting the perturbation-based strategy is to show that improving robustness may contribute to fast generalization, thus further validating our main theorem. This strategy is mainly used to validate the theory and we do not aim to show advantage to data augmentation methods. And we discussed in our paper that in some cases, data augmentation can be seen as a sort of perturbation and may explain why grokking does not happen in standard cases when augmentations are employed. The main relationship is that the perturb strategy can be seen as a special case of augmentation, but it is just one of the ways we consider that can easily enhance robustness and not the only way. And also we are the first to employ this strategy in the grokking literature.
>
>
>
> >Q3: Although the decay in the l2 weight norm is related to grokking, how does this phenomenon relate to the model’s learning dynamics and generalization ability? This relationship could be more complex in larger models. Can the authors’ theoretical explanation be extended to more diverse model architectures and tasks?
>
> A: The decay of $l_2$ weight norm may correlate with the model's generalization in a somewhat complex manner [1].  Our paper's theoretical explanation does not rely on specific model architectures as long as it satisfies the basic settings the theorem requires. When tasks and datasets are more complex, it may violate the assumption of the theorem, for e.g. the training sets may not be fitted well in some complex cases.
>
> [1] A PAC-BAYESIAN APPROACH TO SPECTRALLY-NORMALIZED MARGIN BOUNDS FOR NEURAL NETWORKS

---

> ### Author Response · Authors · 2024-11-25
> **A request for feedback**
>
> We sincerely appreciate your valuable time devoted to reviewing our manuscript. ***We would like to gently remind you of the approaching deadline for the discussion phase.*** We have diligently addressed the issues you raised, providing detailed explanations. Given the importance of your feedback in refining and improving the work, we would greatly appreciate it if you could review the rebuttal at your earliest convenience.

---

### Official Review · Reviewer_7aQL · 2024-11-04

**Soundness:** 3
**Presentation:** 3
**Contribution:** 3
**Rating:** 5
**Confidence:** 3

**Summary:**

The paper shows that the L2 weight norm decay is connected to generalization and designs an adaptive data augmentation method to speed up generalization

**Strengths:**

- The paper studies how to speed up generation in grokking, that is an important problem.
- The theory seems to match the experiments well.

**Weaknesses:**

In Figure 4, how are the hyperparameters (learning rate, weight decay) selected? In the standard training of Figure 4b, the generalization speed may be much faster with proper tuning. It would make the generalization speed-up method more convincing by carefully tuning the hyperparams.

**Questions:**

- Where did you define $a$ and $b$ shown in Corollary 4.3? The only place a,b are defined is Sec 3, where (a,b) represents the input pair.
- Why do you set L=1/2, a=1925, b=500? Does Corollary 4.3's predicted accuracy change with learning rate and weight decay?
- [1] mentioned that there is almost no grokking phenomenon for the MNIST experiments under standard training, and they induce clear grokking by setting large initialization scale. But Figure 2a and 2c show clear grokking for MNIST. What initialization scale is used in those experiments?

[1] OMNIGROK: GROKKING BEYOND ALGORITHMIC DATA

---

> ### Author Response · Authors · 2024-11-23
> **Rebuttal to Reviewer 7aQL**
>
> >W1: In Figure 4, how are the hyperparameters (learning rate, weight decay) selected? In the standard training of Figure 4b, the generalization speed may be much faster with proper tuning. It would make the generalization speed-up method more convincing by carefully tuning the hyperparams.
>
> A: In Figure 4, the hyperparameters are all selected based on prior work [1], which we clearly stated in the preliminary section (section 3). And in Figure 4 (b), the standard training and perturb training share the same hyperparameters (like learning rate, weight decay) to make the comparison fair.
>
> [1] OMNIGROK: GROKKING BEYOND ALGORITHMIC DATA
>
> >Q1: Where did you define a and b shown in Corollary 4.3? The only place a,b are defined is Sec 3, where (a,b) represents the input pair.
>
> A: The a and b are hyperparameters and are just defined in the context of corollary 4.3, sorry for the abuse of notation.
>
> >Q2: Why do you set L=1/2, a=1925, b=500? Does Corollary 4.3's predicted accuracy change with learning rate and weight decay?
>
> A: We set these hyperparameters so that the calculated predicted accuracy can match the empirically observed accuracy well. The predicted accuracy may change with learning rate and weight decay as these may affect the learning dynamics and thus affect the suitable parameters a and b from the defined equation of hyperparameters a and b.
>
> >Q3: [1] mentioned that there is almost no grokking phenomenon for the MNIST experiments under standard training, and they induce clear grokking by setting large initialization scale. But Figure 2a and 2c show clear grokking for MNIST. What initialization scale is used in those experiments?
>
> A: Thank you for your question and careful reading. We adopt the same setting as [1] and setting large initialization scale.

---

> > ### Comment · Reviewer_7aQL · 2024-11-25
> >
> > Thank you for your reply. I still think the Perturbing training curve looks unusual in Figure 4b. It first quickly rises to around 30% test accuracy. Notably, the training data is also 30% of the full dataset. A conjecture is that you did not add positional embedding in the transformer, so that the model cannot tell pair (a,b) from (b,a), so you can quickly generalize on test data (b,a) if (a,b) is in the training set. However, this phenomenon did not occur in your standard training curve. Do you have any intuitive explanation for this unusual phenomenon?
> >
> > In Corollary 4.3, can you find out some relationship between (L,a,b) and (learning rate, weight decay)? It would be interesting if such a relationship exists.

---

> > > ### Author Response · Authors · 2024-11-25
> > > **Further response**
> > >
> > > Thank you for your question. For the first question, we add learnable position embeddings in the network. But we think your hypothesis is very close to a plausible explanation which we will briefly discuss: We think the reason why it reaches about 30 % is that: It first leans commutative law and making test predictions (b, a) (whose (a, b) in is training set) will be correct (See Figure 6) and the proportion will be about 30 %.
> > >
> > > For the second question, it is hard to analytically give relationships. But some qualitative analysis can be made, for example when weight decay is larger, the weight norm decays more rapidly, making the hyperparameter b increases and a decreases.

---

> > > > ### Comment · Reviewer_7aQL · 2024-11-25
> > > >
> > > > Thank you for your reply. It would be interesting to see if the model can quickly learn the commutative law with noise injection in Figure 4b.
> > > > My last question is why the train and test abelian acc both start from exactly 100% in Figure 5(b), while they are only 80% in Figure 5(a)? This does not seem obvious given the positional embedding are randomly initialized at the beginning.

---

> > > > > ### Author Response · Authors · 2024-12-01
> > > > >
> > > > > For the first question: noise injection in Figure 4 (b)'s setting is the same as in Figure 5 (b), which we can see the model quickly learns commutative law on the training dataset.
> > > > >
> > > > >
> > > > > For the second question: The accuracy plotted in Figure 5 (a) is the test accuracy and test accuracy in the standard training is about 80 %, which is the same as in Figure 1 (a).

---

> ### Author Response · Authors · 2024-11-25
> **A request for feedback**
>
> We sincerely appreciate your valuable time devoted to reviewing our manuscript. ***We would like to gently remind you of the approaching deadline for the discussion phase.*** We have diligently addressed the issues you raised, providing detailed explanations. Given the importance of your feedback in refining and improving the work, we would greatly appreciate it if you could review the rebuttal at your earliest convenience.

---

### Official Review · Reviewer_n5oG · 2024-11-04

**Soundness:** 3
**Presentation:** 2
**Contribution:** 2
**Rating:** 3
**Confidence:** 3

**Summary:**

This work explains grokking, a large accuracy gap between training and test errors in the early stage of the training, with a sharp catch-up of test errors in the late stage. The authors show the importance of weight decay (or small L2 norm of parameters) from the nearest-neighbor perspective (Theorem 4.2). Further, several interesting observations are made empirically in terms of the commutativity and information-theoretic metrics that correlate better with the timing of the grokking.

**Strengths:**

- Grokking is explained from their lower bounds of test accuracy.
- Several interesting observations are made in terms of the commutativity and information-theoretic metrics.

**Weaknesses:**

I raise the following as the major weaknesses of this work.
1. Limited technical contributions
2. Limited justifications and interpretations for the observations
3. Poor paper writing

I elaborate on the weaknesses below.

1. Limited technical contributions
This paper has only one technical claim (Theorem 4.2) with the associated corollary. Theorem 4.1 should be a direct adaptation from a related work. The proof of Theorem 4.2 is also straightforward. Further, it is not very clear to me whether the conditions in Theorem 4.2 to guarantee the lower bound of test accuracy are likely to be satisfied in the realistic datasets and whether it really explains the grokking that occurs in various types of datasets. To me, Theorem 4.2 certainly gives an intuitive but still weak argument.

Corollary 4.3 gives a more concrete lower bound, but it was not very understandable as several definitions are missing (related to Weakness 3). What are $L$, $a$, and $b$? The programming-like variable "train-acc" is also not professional. The condition $\|\mathbf{x}\|_2 = 1$ seems to assume one-hot encoding of a number or a concatenation of two encoding vectors, but no details are given.

Figure 3 seems amazingly fits to the empirical test accuracy. However, the choice of the values of $(L, \mu, a, b)$ is not justified. To me, these values appear to be carefully selected for the fitting.

2. Limited justifications and interpretations for the observations
The observations on the commutativity and the better correlating metrics are interesting. It would be better if the authors could justify them theoretically or at least give reasonable interpretation. As for the metrics, I don't see why these better correlate with the timing of grokking - is it trivial or not - and their utility.

2. Poor paper writing
As partially given in Weakness 1, this paper does not provide sufficient details of the symbols and setup. To list a few,
- [Sec. 3] The encoding of numbers is not given. The theoretical model of the neural network is not given (ReLU network?). Numerical experiments are done with transformer models, but the theory is for MLPs. What is the one-layer ReLU transformer? (one encoder layer or decoder layer)?
- [Sec. 4.1, line 158] $\mathrm{onehot}(\,\cdot\,)$ is not defined.
- [Corollaly 4.3] See Weakness 1.

**Minor comments**
- [Sec 5.2, line 322] "(a) shows that ... but (b) [Perturb learning?] has a chaotic."

**Questions:**

Please answer the two weaknesses raised above.

---

> ### Author Response · Authors · 2024-11-23
> **Rebuttal to Reviewer n5oG**
>
> >W1: Limited technical contributions
>
> A: The theorem is based on a lemma from literature, but the point is we are the first to derive a more suited theorem in the grokking literature that explains the role of weight norm decay. And the assumption we made are very mild and are very common among literature. What's more, the paper is not just about this theorem, we also provide many other contributions like the degrokking strategy and metrics and thus study the grokking phenomenon in depth.
>
> >W2: Limited justifications and interpretations for the observations
>
> A: It is hard to provide a theorem of why perturb training will result in faster commutative law. But we may have an intuitive explanation as follows: Because of perturb training, when input pair (a, b)'s representation is perturbed a bit the prediction will still remain true. As the representation of (a, b) and (b, a) is much similar than the similarity of (a, b) and other (a', b'), then the prediction of pair (a, b) may become same for the prediction of (b, a), making the commutative law becomes valid. For the better correlation of metrics, mainly conclude this from an inspection of Figures. For example, in Figure 8, the metric changes sharply when test accuracy groks which we also marked in the picture and discussed in the respective section.
>
>
> >W3: Poor paper writing
>
> A: Thank you for your reading, we may will the symbols more clearly to further improve readability. And the answers to your questions are as follows:
>
> (a) The ReLU transformer is a transformer with a ReLU activation function. One layer is the standard meaning of one attention block and MLP block.
>
> (b): Onehot(y) notation is a standard notation that represents a vector with only the y-th coordinate 1 and others 0.
>
> (c): a and b are hyperparameters just defined in the context of corollary 4.3 and L is defined in theorem 4.2 as the lip constant. The use of abbreviations like train-acc is because that the name in full will make the picture ugly and not clear.

---

> > ### Comment · Reviewer_n5oG · 2024-11-25
> >
> > I thank the authors for the response.
> >
> > In the response to (W1),  the authors claims that their contributions include several empirical approaches (e.g., degrokking), but as I mentioned in (W2), there are little understanding and justification about them. The parameter choice in Figure 3 is also not well explained.
> >
> > Overall, I don't feel my concerns are addressed sufficiently and thus would like to retain current evaluation.

---

> > > ### Author Response · Authors · 2024-11-25
> > > **Further response**
> > >
> > > Thank you for your reply. For your comments on "understanding and justification", we have already answered that in our previous rebuttal on W2. If you still have something unclear, please let us know and we will present more details. And for Figure 3, the basic idea for choosing parameters is to show that under some tendency assumptions, the estimated probability bound can approximately match the real test accuracy, it can be seen as applying theorem 4.2's bound to make it much more "useful".

---

> ### Author Response · Authors · 2024-11-25
> **A request for feedback**
>
> We sincerely appreciate your valuable time devoted to reviewing our manuscript. ***We would like to gently remind you of the approaching deadline for the discussion phase.*** We have diligently addressed the issues you raised, providing detailed explanations. Given the importance of your feedback in refining and improving the work, we would greatly appreciate it if you could review the rebuttal at your earliest convenience.

---

### Official Review · Reviewer_evAa · 2024-11-06

**Soundness:** 2
**Presentation:** 1
**Contribution:** 3
**Rating:** 5
**Confidence:** 4

**Summary:**

The paper aims to understand the grokking phenomenon deeper and proposes a method to mitigate grokking to achieve faster generalization. Authors provide both theoretical and experimental evidences for the grokking phenomenon and shows their perturbation method accelerates the generalization speed. Authors also provide 2 metrics PMI and PE that aim to better monitor when grokking happens.

**Strengths:**

The novelty and contribution of this paper is good, focusing on grokking problem, authors provide better metrics and algorithm to accelerate model generalization.
The paper has both theoretical and empirical results.

**Weaknesses:**

1. Figures' x-axis switch between logarithmic and linear, make it very hard to do a comparison among figures. Especially in Figure 3, curves are compressed by logarithmic x-axis on larger step numbers make it very difficult to understand the difference.
2. It is vague that how authors choose the transition point in all figures, for example, in Fig 1(a), the transition point for weight norm is selected at the beginning of weight norm start decrease, but in Fig 9(b), where PE has a similar curve, the transition point is selected much later to convey the point that PE is more aligned with Test Acc. The vague selection also happens in 8(b) 9(a) 10(b) 14(a) and etc. I suggest authors provide a consistent principle in choosing those points to reduce bias in result presenting.
3. I didn't really get the definition of PMI, in ln 394, $W_1$ is the first layer so the $W_2$ is the second layer? Is the PMI defined as the layer-wise mutual information on feature embedding gram matrix? also, in ln 399, It is not clear to my that $z^{(2)}_i$ is defined on whether entire $W$ or $W_2$?
4. Results and experiments are mostly based on shallow network and no deep network results.
5. Combine 3 and 4, if I understand correctly PMI and PE will be extremely costly when network being larger and deeper, what will be the complexity of PMI and PE regarding the number of layers and representation dimensions?
6. Ln 429 states that in Fig. 9, the PE changes sharply but 9(a) doesn't, and 9(b) also changes earlier than test acc (see weakness 2).

**Questions:**

1. What is meaning of X-axis in figure 1 and 2? There are several more figures don't have x-axis labels.
2. What is Algo refer to in Fig 2?
3. In Fig. 7, the logit distance has a significant scale difference(0-15000 vs. 0-150), does standard training also has the "chaotic behavior" if we zoom-in?

---

> ### Author Response · Authors · 2024-11-23
> **Rebuttal to Reviewer evAa**
>
> >W1: Figures' x-axis switch between logarithmic and linear, make it very hard to do a comparison among figures. Especially in Figure 3, curves are compressed by logarithmic x-axis on larger step numbers make it very difficult to understand the difference.
>
> A: We sometimes switch between logarithmic and linear to make the plotted pattern more visible because in cases log axis may make the changing pattern not very clear. And in Figure 3, ***there may be a misunderstanding, the difference between the two curves is small is just what we needed as we want the predicted accuracy to match the real one.***
>
>
> >W2: It is vague that how authors choose the transition point in all figures, for example, in Fig 1(a), the transition point for weight norm is selected at the beginning of weight norm start decrease, but in Fig 9(b), where PE has a similar curve, the transition point is selected much later to convey the point that PE is more aligned with Test Acc. The vague selection also happens in 8(b) 9(a) 10(b) 14(a) and etc. I suggest authors provide a consistent principle in choosing those points to reduce bias in result presenting.
>
> A: ***There may be a misunderstanding. Figure 1 and other Figures are different.*** As we pointed in the paper, Figure 1's weight norm decays before the test accuracy changes, while the other proposed metrics correlate more closely with test accuracy change. Thus the transition point in Figure 1 is different as it does not align with the transition of test accuracy.
>
>
> >W3: I didn't really get the definition of PMI, in ln 394, $W_1$ is the first layer so the $W_2$ is the second layer? Is the PMI defined as the layer-wise mutual information on feature embedding gram matrix? also, in ln 399, It is not clear to my that $z^{(2)}_i$ is defined on whether entire $W$ or $W_2$?
>
> A: $W_1$ is the first layer parameter and $W_2$ are the parameters of network except the first layer. PMI can be understood as mutual information between the output of first layer and final output layer. $z^{(2)}_i$ is defined on entire $W$.
>
> >W4: Results and experiments are mostly based on shallow network and no deep network results.
>
> A: The adoption of small-scale tasks and datasets in our study follows the research paradigm in existing literature, as related works focus on small-scale settings. This limitation has been noted in our conclusion section.
>
> >W5: Combine 3 and 4, if I understand correctly PMI and PE will be extremely costly when network being larger and deeper, what will be the complexity of PMI and PE regarding the number of layers and representation dimensions?
>
> A: The complexity will increase as the network goes deeper, as the computation relies on the output of the network, but the cost is not that big as the inference speed on GPU is quite fast. The cost reliance on dimension is basically on the calculation of Gram matrix, which is also quite efficient on GPU.
>
> >W6: Ln 429 states that in Fig. 9, the PE changes sharply but 9(a) doesn't, and 9(b) also changes earlier than test acc (see weakness 2).
>
> A: We think the changing pattern can be seen as sort of evident when task accuracy drastically changes, thus making PE valuable. But of course, the criteria of whether it is timely or not is somewhat subjective and we offer this metric as it is theoretically nuanced and may have some value in our own opinion.
>
>
> >Q1: What is meaning of X-axis in figure 1 and 2? There are several more figures don't have x-axis labels.
>
> A: It means the number of optimization steps.
>
> >Q2: What is Algo refer to in Fig 2?
>
> A: As we stated in the caption of this Figure. we use Algo to indicate Modulo Addition dataset to save the space for image caption.
>
> >Q3: In Fig. 7, the logit distance has a significant scale difference(0-15000 vs. 0-150), does standard training also has the "chaotic behavior" if we zoom-in?
>
> A: No, the case of standard training will have the distance nearly 0.

---

> > ### Comment · Reviewer_evAa · 2024-11-26
> >
> > W1: I understand what authors mean "the difference between the two curves is small is just what we needed as we want the predicted accuracy to match the real one", however, Authors use 3/4 of the figure on 1-100 steps where the accuracy is almost 0. Then the log scale on x axis squeeze the 100 to 1000 and only occupy 1/4 of the figure, where the actual accuracy increase happened.  This way of experiment presentation will hide the details in 100-1000 steps,  to be specific, a small difference in the last several ticks may result a larger difference in linear scale of steps.
> >
> > W2: Authors may misunderstand my question, I'm trying to understand the criteria of choosing the small square markers on the curve. Fig 1(a) put the square marker of weight norm much earlier compared to Fig 9(b) of PE curve, and both curves have similar shape. Why do authors think Fig 9(b) PE curve should have square marker on later steps? What is the criteria on choosing the marker position? If the choice of marker is not consistent, it will potentially introduce reporting bias. 8(b) PMI, 9(a) PE, 13(b), 15(a)(b) also have the similar marker selection questions.
> >
> > The presentation score were mostly came from the above 2 weakness which may have potential reporting bias. If authors could provide further evidence to share I'd like to adjust my score.
> >
> > Q1: The updated manuscript still has several figures missing x-label, although I currently know its meaning, it would be better to add them for clarity.

---

> > > ### Author Response · Authors · 2024-12-01
> > >
> > > For your first question: For the comparison Figure in the normal scale, please see https://postimg.cc/0KzS9JYS. We think the difference is not significant and holds similar trends with the log scale picture.
> > >
> > >
> > > For your second question: The criteria is mostly based on selecting the point when the indicator metric changes dramatically or has a notable different trend in the curve. It can be roughly identified by noticing the point where the first order derivative value changes.
> > >
> > >
> > > For your third question: Thank you for your comment. We will adjust this accordingly to make it clearer.

---

> ### Author Response · Authors · 2024-11-25
> **A request for feedback**
>
> We sincerely appreciate your valuable time devoted to reviewing our manuscript. ***We would like to gently remind you of the approaching deadline for the discussion phase.*** We have diligently addressed the issues you raised, providing detailed explanations. Given the importance of your feedback in refining and improving the work, we would greatly appreciate it if you could review the rebuttal at your earliest convenience.

---

### Meta-Review · Area_Chair_s6zT · 2024-12-21

**Metareview:**

This paper summarizes a recent study on the phenomenon of grokking, where neural networks exhibit generalization long after overfitting the training data. The authors investigate this phenomenon from a robustness perspective, suggesting that the commonly observed decrease in neural network weight norm is theoretically linked to grokking. They propose perturbation-based methods to enhance robustness and accelerate the generalization process. Additionally, the study explains the speed-up in generalization by showing its connection to the model learning the commutative law, a key condition for grokking on test data. The authors further observe that weight norm correlates with grokking inconsistently over time and introduce new robustness-based metrics that better capture this phenomenon.

According to the reviewers the strengths and weaknesses of the paper are as follows:

Strengths:

+The paper provides a fresh perspective on the grokking phenomenon, connecting it with robustness and weight norm decay, offering both theoretical and empirical insights.

+The paper proposes a perturbation-based approach to enhance robustness and accelerate generalization, which is novel in the context of grokking.

+The paper introduces new metrics (PMI and PE) that better correlate with grokking, supported by empirical validation.

+Theorem 4.2 establishes a connection between weight decay and test accuracy, aligning theory with empirical results.

+Empirical results, though limited in scope, support the proposed theory and demonstrate the utility of the new metrics.

Weaknesses:

-Presentation Issues: Figures lack consistent axis scaling (e.g., switching between logarithmic and linear scales) and clear labeling, making them hard to interpret. Marker selection for transition points appears inconsistent, leading to potential bias.

-Limited Generality: The study is based on small-scale tasks and datasets, making it difficult to generalize findings to larger, more complex models or datasets.

-Weakness in Theoretical Depth: The technical contributions, particularly Theorem 4.2, are considered straightforward and not fully justified for practical datasets. The proposed metrics, while empirically supported, lack theoretical backing.

-Comparison to Standard Methods: The perturbation-based approach is not clearly differentiated from traditional data augmentation methods, raising questions about its unique advantages.

-Writing and Clarity: Symbols, notations, and experimental setups are inadequately explained, reducing the accessibility and readability of the work.

The authors in their rebuttal defended the choice of metrics and marker points but acknowledged the need for clearer presentation. The also justified the small-scale focus as consistent with prior literature but noted this limitation in the conclusion. Finally, they also explained the rationale for hyperparameter choices in experiments, though reviewers found these explanations insufficient.

While acknowledging the novelty and potential of the work, reviewers remained unconvinced about its broader applicability and clarity.
Suggestions for improvement included theoretical justifications for metrics, consistent figure formatting, and extending the approach to more complex scenarios. Overall, while the paper introduces interesting ideas I concur with the reviewers and do not think the paper is ready for publication in its current form. However, I encourage the authors to revise and resubmit the paper to a future venue.

**Additional Comments On Reviewer Discussion:**

While acknowledging the novelty and potential of the work, reviewers remained unconvinced about its broader applicability and clarity.
Suggestions for improvement included theoretical justifications for metrics, consistent figure formatting, and extending the approach to more complex scenarios. Overall, while the paper introduces interesting ideas I concur with the reviewers and do not think the paper is ready in its current form. However, I encourage the authors to revise and resubmit the paper to a future venue.

---

### Decision · Program_Chairs · 2025-01-22

Reject